# LARP7 family proteins have conserved function in telomerase assembly

Laura C. Collopy[1], Tracy L. Ware[1,2], Tomas Goncalves[1,3], Sunnvør í Kongsstovu[1,4], Qian Yang[1], Hanna Amelina[1], Corinne Pinder[1,3], Ala Alenazi[1,4], Vera Moiseeva[1], Siân R. Pearson[1], Christine A. Armstrong[1] & Kazunori Tomita [1]

Understanding the intricacies of telomerase regulation is crucial due to the potential health benefits of modifying its activity. Telomerase is composed of an RNA component and reverse transcriptase. However, additional factors required during biogenesis vary between species. Here we have identified fission yeast Lar7 as a member of the conserved LARP7 family, which includes the *Tetrahymena* telomerase-binding protein p65 and human LARP7. We show that Lar7 has conserved RNA-recognition motifs, which bind telomerase RNA to protect it from exosomal degradation. In addition, Lar7 is required to stabilise the association of telomerase RNA with the protective complex LSm2–8, and telomerase reverse transcriptase. Lar7 remains a component of the mature telomerase complex and is required for telomerase localisation to the telomere. Collectively, we demonstrate that Lar7 is a crucial player in fission yeast telomerase biogenesis, similarly to p65 in *Tetrahymena*, and highlight the LARP7 family as a conserved factor in telomere maintenance.

[1] Chromosome Maintenance Group, UCL Cancer Institute, University College London, London WC1E 6DD, UK. [2] Department of Biology, Salem State University, Salem, MA 01970, USA. [3] Division of Biosciences, Faculty of Life Sciences, University College London, London WC1E 6BT, UK. [4] MSc Human Molecular Genetics, Faculty of Medicine, Imperial College London, London SW7 2AZ, UK. Correspondence and requests for materials should be addressed to K.T. (email: k.tomita@ucl.ac.uk)

Telomerase is an RNA-containing reverse transcriptase that copies new telomeric repeats onto chromosome ends and functions to maintain cell division[1]. Telomerase is expressed in stem cells, germ cells and unicellular organisms[2,3]. Conversely, most somatic cells have very low telomerase activity, although this is reactivated in ~90% of cancers[4]. Telomerase activity is therefore tightly regulated to meet the needs of the cell. A number of premature ageing diseases, including the bone marrow failure syndromes, dyskeratosis congenita and aplastic anaemia, are caused by genetic mutations in genes encoding factors involved in telomerase biogenesis and activity[5]. These patients have a deficiency in telomerase activity/stability, causing critically short telomeres. This is particularly threatening to cells with a high turnover, such blood progenitor cells. Disease in 30–40% of these patients remains uncharacterised at the genetic level[6]. Thus, it is this imperative to gain an understanding of the telomerase biogenesis pathway.

In humans, telomerase is composed of a reverse transcriptase (hTERT), which uses the RNA component (hTERC) to dock onto the 3′ single-stranded telomere end. hTERT may then processively synthesise telomeric repeats from the template provided by hTERC, before dissociating[7–9]. All telomerase RNAs possess a 3′ end element necessary for its stability[10]. In hTERC, this is two stem-loop structures separated by an H-box (ANANNA) and ACA motif (H/ACA). The binding of telomerase factors dyskerin, NOP10, and NHP2 at the H/ACA motif form the so-called 'pre-ribonucleo-protein complex', before GAR1 binds in transition to the mature RNP[11,12]. hTERC then binds to chaperone TCAB1, which assists its trafficking to the Cajal bodies where the functional telomerase complex localises[13]. Recruitment to the telomeres in S-phase is mediated by the protective complex shelterin[14,15]. Correct assembly of the telomerase complex, with appropriate co-factors for maturation, stability, and subcellular localisation, is necessary for its function and thus telomere maintenance.

Whereas telomerase RNA is functionally conserved, the RNA structure, maturation process, and the RNA-binding co-factors are diverse among species[16]. In fission yeast *Schizosaccharomyces pombe*, the telomerase RNA gene *ter1*+ contains an intron flanked by two exons[17,18]. Following transcription, like hTERC, TER1 precursor is also polyadenylated. The 3′ end of exon 1 has a binding site for Sm family proteins. Sm and Sm-like proteins (LSm) are evolutionarily conserved families that function broadly in the processing of RNA. Seven Sm proteins (SmB1, SmD1, 2 and 3, SmE, SmF and Smz) attach at this site in the poly-adenylated TER1 in a characteristic 'wheel' conformation, stimulating a single spliceosomal cleavage event[19]. However, instead of a second cleavage followed by exon ligation, exon 1 is released[17]. The Sm complex is replaced with the LSm2–8 complex, during which time exonucleolytic cleavage trims the 3′-end to expose a poly-uracil sequence[19]. Once the *S. pombe* telomerase complex has assembled, it is recruited to the telomeres by Ccq1 and Tpz1 (equivalent to human ACD/TPP1), components of the *S. pombe* shelterin complex, similarly to humans[20–24]. Telomerase is then enzymatically active following stable association with the telomere-binding proteins.

In order to examine the factors involved in regulating telomere length homoeostasis, Liu et al.[25] carried out a genome-wide screen of deletion mutations that impacted telomere length in *S. pombe*[25]. Deletion of 168 genes was found to alter telomere length, with four causing 'very short' telomeres (>150 bp shorter than wild type). These were *est1*, *trt1* and *ccq1*, which are essential for telomerase activity, and *pof8*, of unknown function. Pof8 is a putative F-box protein, which bind to Skp1 and Cullin to form an SCF E3 ubiquitin ligase complex[26]. Previous studies identified Pof8 as a Skp1 binding protein but the association with Cullin was not determined[27,28], implying a function distinct from a canonical F-box protein. Here we redefine Pof8 as an RNA-binding protein of the ancient LARP7 family (La

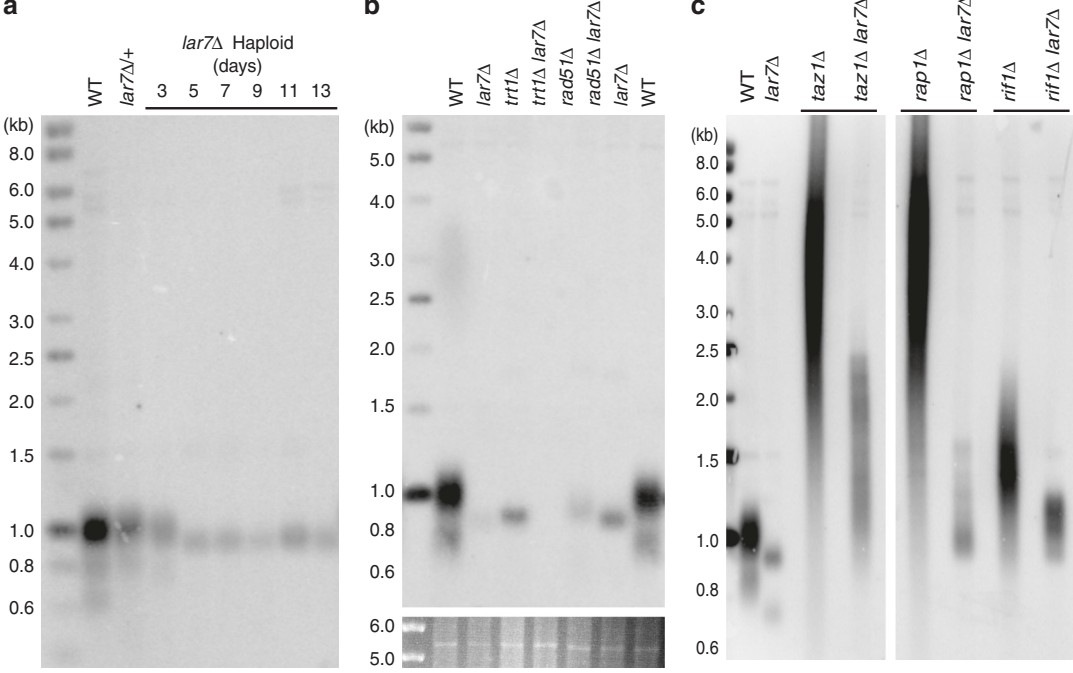

**Fig. 1** *lar7Δ* impairs telomerase activity. **a**–**c** Telomere southern blots of genomic DNA digested with *Eco*RI and hybridised with a telomeric probe. **a** Cells heterozygous for *lar7Δ* retained telomeres of wild-type length. Haploid *lar7Δ* cells were passaged every other day post sporulation and genomic DNA collected at the days stated. **b** The *trt1* and *rad51* genes were deleted in wild-type and *lar7Δ* strains and passaged for a week. Genomic DNA was collected and subjected to telomere southern blot. **c** The *lar7* gene was deleted in the *taz1Δ*, *rap1Δ* and *rif1Δ* strains and cells were passaged for a week. Genomic DNA was collected and subjected to telomere southern blot

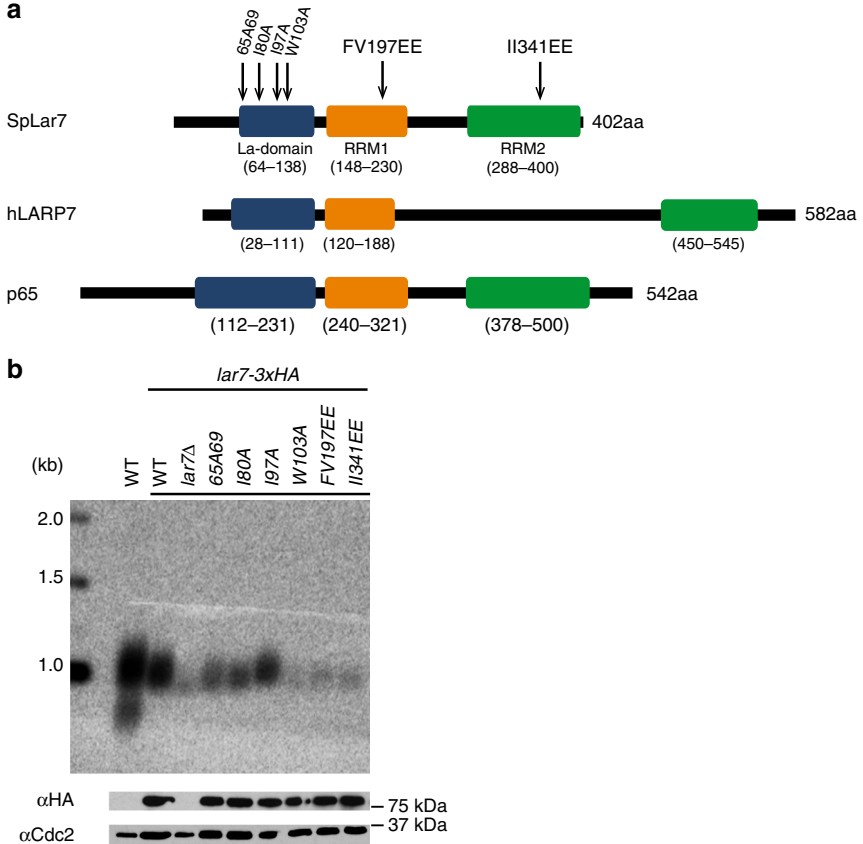

**Fig. 2** Lar7 is an RNA-binding protein of the LARP7 family. **a** Schematic representation of the predicted La-domain and RNA-recognition motifs (RRMs) in *S. pombe* Lar7. Amino acid numbers of each domain are stated in brackets. Strains with the Lar7 point mutations indicated were made for further study. Lar7 is aligned with hLARP7 and p65 based on a position of each domain. **b** Telomere southern blot of six *lar7* mutant strains. Western blot (probing with α-HA) confirms stability of each mutant protein. Original blot image is shown in Supplementary Fig. 4. Cdc2 was used to control for sample input

ribonucleoprotein domain family member 7 or La-related protein 7) and thus rename it Lar7. LARP7 proteins are characterised by an N-terminal RNA-binding domain termed a La-motif and two RNA-recognition motifs (RRMs). LARP7 proteins associate with non-coding RNAs transcribed by RNA polymerase III, which have a hallmark 3′-UUU-OH tag[29]. Here we report that Lar7 interacts with TER1 and plays a crucial role during its biogenesis. Lar7 stabilises the interaction between TER1 and the Lsm2–8 complex and facilitates assembly with Trt1. Loss of Lar7 leads to disassembly of telomerase and exosomal degradation of TER1, resulting in impaired telomerase activity and maintenance of very short telomeres. Further, we report a high degree of conservation between Lar7 and both the human protein hLARP7 and *Tetrahymena thermophila* telomerase RNA-binding protein p65. In human cells, loss of hLARP7 has recently been shown to cause telomere shortening[30]. Thus, LARP7-familiy proteins are universally crucial for telomere maintenance.

## Results

**lar7Δ cells have reduced telomerase activity.** To investigate the cause of short telomeres in the absence of Lar7 (originally identified as Pof8), a haploid *lar7Δ* deletion strain was generated from a heterozygous diploid parent, and passaged over generations. Telomere lengths were measured by southern blot and found to be considerably shorter than wild-type cells by day 5 (Fig. 1a). Such a generation-associated shortening of telomeres suggests loss of telomerase activity. However, critically short telomeres were maintained over multiple generations. Following the deletion of *trt1⁺* but not *rad51⁺*, which is required for homologous

recombination, the short telomeres in *lar7Δ* were lost (Fig. 1b). Thus, *lar7Δ* cells maintain short telomeres using telomerase and not via an alternative pathway.

Deletion of genes encoding the telomere-associated proteins, *taz1*, *rap1* or *rif1*, leads to deregulated recruitment of telomerase and, consequently, telomerase-dependent telomere elongation[21,31,32]. In double mutants with *lar7Δ*, such telomere elongation was diminished (Fig. 1c). The expression of the telomerase and shelterin protein components that modulate telomerase activity was unaffected in *lar7Δ* cells (Supplementary Fig. 1). Thus, shortening of telomeres in *lar7Δ* cells is not due to impaired stability of telomeric proteins, rather telomerase activity itself is reduced in the absence of Lar7.

**Lar7 is a member of the LARP7 RNA-binding protein family.** In order to assess the function of Lar7 in telomere biology, we first analysed its predicted structure for key domains or moieties that may allude to its function. Using the bioinformatics tool HHpred (homology detection and structure prediction by HMM-HMM) and the Protein Homology/Analogy Recognition Engine v2.0 (Phyre2)[33,34], we identified two likely RNA-recognition motifs (RRMs), which have the format β1-α1-β2-β3-α2-β4, in four fission yeast species (Supplementary Fig. 2). Additionally, we identified structural similarity between Lar7 and the human protein hLARP7 (La-related protein 7 or La ribonucleoprotein domain family member 7), which is part of the wider LARP7 family of non-coding RNA-binding proteins[35]. Remarkably, a well-described member of this family and orthologue of hLARP7 is the telomerase-binding protein p65 in ciliate *T. thermophila*.

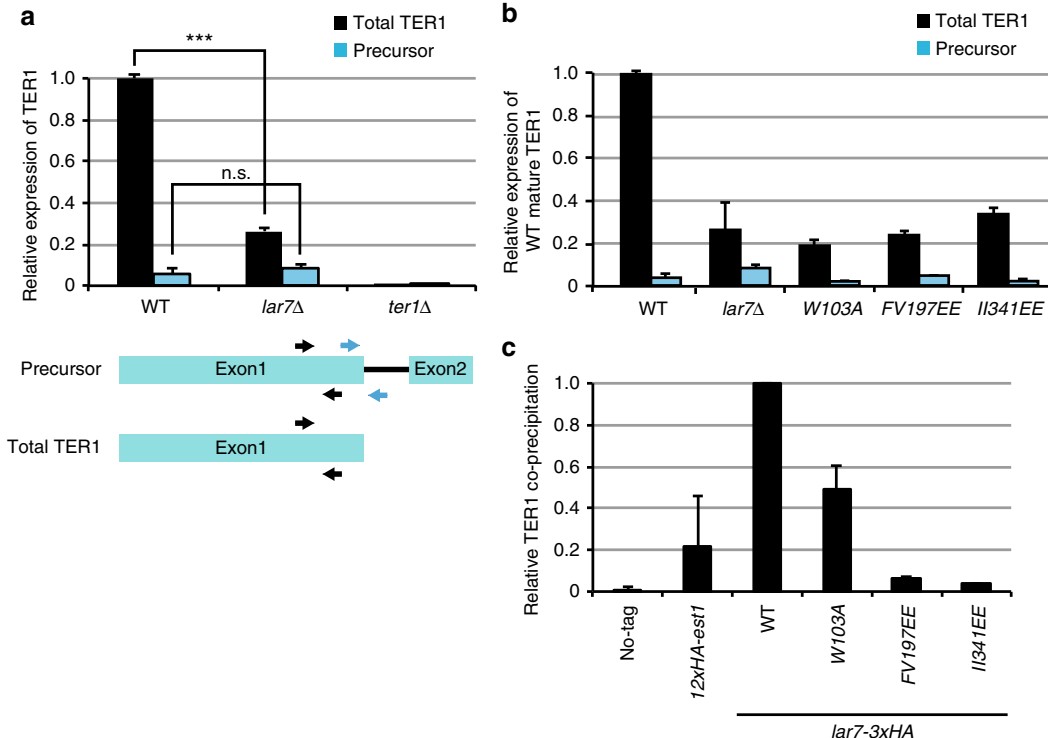

**Fig. 3** Lar7 binds and is required for accumulation of mature TER1. TER1 precursor and/or mature mRNA was quantified using RT-qPCR and normalised first to actin and then to the expression in wild-type cells. **a** Positions of the primers specific to precursor (blue) and total (black) TER1 are indicated. Significantly less total TER1 expression was detected in *lar7Δ* compared to wild-type (unpaired *t* test *** at *p* < 0.001). Data represented as a mean of three independent experiments. Error bars show standard deviation. **b** Quantification of mature TER1 in three *lar7* point-mutant strains. **c** Lar7 interacts with TER1 via its RNA-recognition motifs. Presence of TER1 mRNA in lysates following immunoprecipitation using α-HA was measured using reverse-transcriptase quantitative polymerase chain reaction (RT-qPCR) and normalised first to actin mRNA and expressed as immunoprecipitated RNA over total RNA. The efficiency of RNA precipitation in *lar7* mutants is represented as a percentage of enrichment relative to wild-type Lar7-3xHA. The TER1-binding protein Est1 was used as a positive control. Data is represented as a mean of two independent experiments. Error bars show standard deviation

All LARP-family proteins (including hLARP7 and p65) have a conserved N-terminal La-motif followed by an RRM (together termed the 'La-module') and a second RRM toward the C-terminus (Fig. 2a). The alignment of *S. pombe* Lar7 with p65 and hLARP7 and a comparison of Lar7 with the published structures of hLARP7 and p65[36–38] revealed considerable sequence and structural similarity between the three proteins (Supplementary Fig. 3). The reported secondary structure of a La-motif is α1-β1-α2-α3-β2-β3[39]. According to the HHpred analysis, the N-terminus of *S. pombe* Lar7 corresponds to this predicted structure (amino acids 64–138), preceding RRM1 (amino acids 148–230). We have therefore assigned this region a La-motif. Interestingly, the domain assigned as an F-box in Lar7 (amino acids 66–106[27,28]) lies within the La-motif. Due to this striking conservation and the criteria defining LARP7 family proteins[29], in addition to the function of Lar7 in telomerase RNA binding (defined in this report), we concluded that the annotated Pof8 is a member of the LARP7 family and accordingly renamed it Lar7 (La-related protein 7).

To assess whether the La-motif and the RRMs identified in Lar7 were required for telomere maintenance, a number of conserved residues within the domains were mutated (Fig. 2a and Supplementary Fig. 2). Four target residues within the La-motif were substituted to alanine. Within each RRM, two highly conserved residues on the so-called RNP-1 consensus sequence (β-strand 3) were identified and substituted to glutamate. Lar7 was endogenously tagged with three tandem HA epitopes and we found that the each mutation generated in this study did not impair stability of the Lar7 protein (Fig. 2b and Supplementary

Fig. 4). However, southern blotting in each case revealed short telomeres; especially La-*W103A*, RRM1-*FV197EE* and RRM2-*II341EE* mutations, which confer critically short telomeres, akin to *lar7* deletion. Thus, our mutagenesis analysis suggested that the La-motif and RRMs are necessary for Lar7 function in telomere maintenance (Fig. 2b).

**Lar7 associates with TER1 and prevents exosomal degradation.** As mutations within the Lar7 RRMs caused short telomeres, we investigated whether expression of TER1 was impaired by the loss of Lar7. Once precursor TER1 is transcribed, spliceosomal activity trims the product to release only exon 1, which becomes the mature TER1[19]. Reverse-transcriptase quantitative PCR (qPCR) was carried out to quantify the relative amount of precursor vs. total TER1. This revealed a similar level of precursor RNA between *lar7Δ* cells and wild type, but a significant reduction in the total product in the absence of Lar7 (Fig. 3a), suggesting low expression of the mature form. A reduction in TER1 was also seen in the La-motif mutant *W103A* and the RRM mutants *FV197EE* and *II341EE* (Fig. 3b). As the level of precursor RNA was not reduced in *lar7Δ* cells, we predicted that the stability of the mature TER1 was defective.

To determine if indeed Lar7 binds to RNA, RNA-immunoprecipitation was performed. We found that TER1 co-precipitated with Lar7 endogenously tagged with HA, with ~700-fold enrichment vs. wild-type cells with untagged Lar7 (Fig. 3c). However, the efficiency of TER1 co-precipitation was dramatically reduced with the RRM motif mutants *FV197EE* and

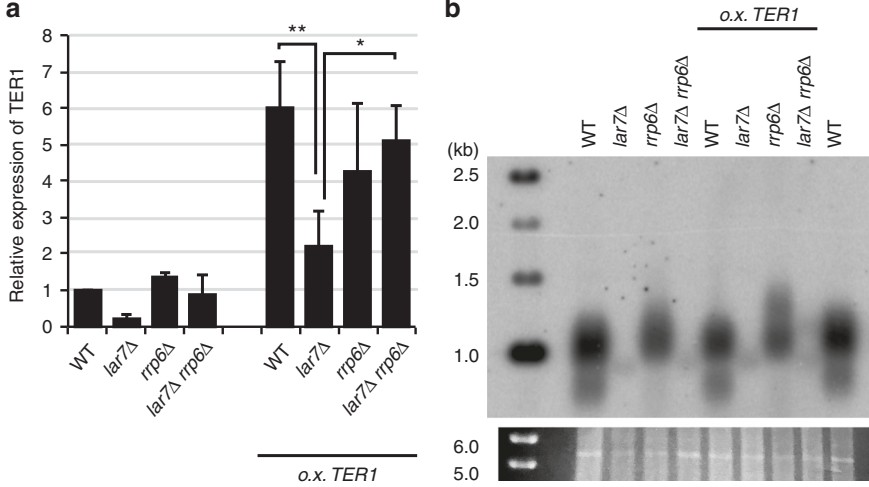

**Fig. 4** *lar7Δ* causes exosomal degradation of TER1. **a** Expression of TER1 mRNA quantified using RT-qPCR and normalised to actin. Expression is shown relative to wild-type cells without the *ter1* overexpression promoter. *rrp6Δ* rescues the reduced expression seen in *lar7Δ* cells. Data represented as a mean of four independent experiments. Error bars show standard deviation. Significantly less TER1 was detected in *lar7Δ* vs WT *ter1* overexpression promoter cells (*p* = 0.0063) and *lar7Δ* vs. *lar7Δrrp6Δ ter1* overexpression promoter (*p* = 0.0242, unpaired *t* test * at *p* > 0.05 and ** at *p* < 0.01). **b** Telomere southern blot. The rescued expression of TER1 in *lar7Δrrp6Δ* cells shown in **a** did not rescue short telomere phenotype (lanes 4 and 8)

*II341EE* and the La-motif mutant *W103A* (Fig. 3c). Thus, the two RRMs and the La-motif of Lar7 are required for the association with TER1 and the interaction between Lar7 and TER1 appears to be crucial for telomere length homoeostasis. We additionally examined if Lar7 interacted with other non-coding RNAs, specifically U1 and U6. We found that neither interacted with Lar7 and loss of Lar7 did not impair their stability (Supplementary Fig. 5). This therefore suggests that Lar7 has an affinity specifically for TER1.

In order to directly address whether loss of mature TER1 in *lar7Δ* cells was associated with impaired transcription or processing of precursor RNA, or with activation of RNA degradation pathways, the endogenous *ter1* promoter was replaced with the *nmt41* overexpression promoter. In the presence of Lar7, the *nmt41*-driven expression led to an approximately sixfold increase in total TER1 levels. However, TER1 expression was significantly reduced in *lar7Δ* cells (Fig. 4a), suggesting that TER1 is actively degraded in the absence of Lar7. The exosome is the main pathway for RNA degradation and is involved in RNA trimming during maturation[40]. The gene *rrp6* encodes for a 3′-5′ exo-ribonuclease subunit of the exosome[41]. The deletion of *rrp6* resulted in the recovery of TER1 expression in *lar7Δ* cells, close to wild-type level (Fig. 4a). A similar recovery was observed with the *nmt41* promoter-expressed TER1. Thus, Lar7 stabilises mature TER1 by inhibiting the exosome Rrp6 activity.

We reasoned that an impaired stability of TER1 in *lar7Δ* cells may be associated with the short telomere phenotype. To this end, we analysed telomere length in the *lar7Δ* cells that expressed a sufficient level of TER1 either by *rrp6Δ* and/or TER1 overexpression. However, despite the rescued levels of TER1, short telomeres were still observed in cells lacking Lar7 (Fig. 4b). Thus, TER1 is not fully functional in the absence of Lar7, even when its degradation is prevented. This suggested that Lar7 has another role in mediating the assembly of the functional telomerase complex.

**Lar7 is required for association of TER1 with LSm and Trt1.** The maturation of precursor TER1 RNA involves the binding of Sm family proteins in order to initiate spliceosomal cleavage and 5′ end capping, followed by LSm2–8 binding to stabilise and process the mature product[19]. We therefore investigated whether these key steps in telomerase biogenesis were defective upon the loss of Lar7. To test this hypothesis, we investigated the association of TER1 with the Sm protein Smb1, LSm component Lsm3 and Trt1. We confirmed that the stability of Smb1, Lsm3 and Trt1 were not impaired in the *lar7* mutants (Fig. 5a and Supplementary Fig. 1). RNA-immunoprecipitation results showed that the amount of TER1 associated with Smb1 was low in both wild-type and *lar7Δ* cells (Fig. 5a). This likely reflects the relatively transient nature of the TER1 interaction with the Sm complex and the relatively low proportion of TER1 precursor[19]. However, following Lsm3 immunoprecipitation, a dramatic reduction of co-precipitated TER1 was observed in the absence of Lar7 (Fig. 5a). Likewise, the level of TER1 that co-precipitated with Trt1 was reduced to approximately 2% of wild-type levels in *lar7Δ* cells. Thus, we concluded that Lar7 is required for telomerase assembly (Fig. 5a).

Co-immunoprecipitation assays confirmed an interaction between Lsm3 and Lar7 (Fig. 5b), indicating that they co-exist as a complex. Their association was sensitive to the presence of RNase and was abolished in *lar7-W103A* and *lar7-FV197EE* mutants. In addition, an interaction between wild-type Lar7 and Trt1 was detected, which was also dependent on RNA (Fig. 5c). These data suggested that Lar7, LSm2–8 and Trt1 independently bind to TER1 to form the telomerase ribonucleoprotein complex. Chromatin immunoprecipitation (ChIP) analysis revealed that telomeric DNA co-purifies with Lar7, suggesting that Lar7 remains bound to TER1 and itself localises to the telomere (Fig. 5d). Further, in *lar7Δ* cells, Trt1 is unable to bind at the telomere, highlighting the need for Lar7 in telomerase assembly and activity at the chromosome ends (Fig. 5d). Collectively, we have demonstrated that Lar7 binding to TER1 facilitates the association with LSm and recruitment of Trt1 to form the mature telomerase complex.

## Discussion
The structure prediction analyses we performed suggested that the previously annotated Pof8 protein is in fact a member of the conserved LARP7 family, which we accordingly named Lar7. Our mutation analysis suggested that the La-motif and two RRMs of Lar7 are all required for TER1 binding and the assembly of

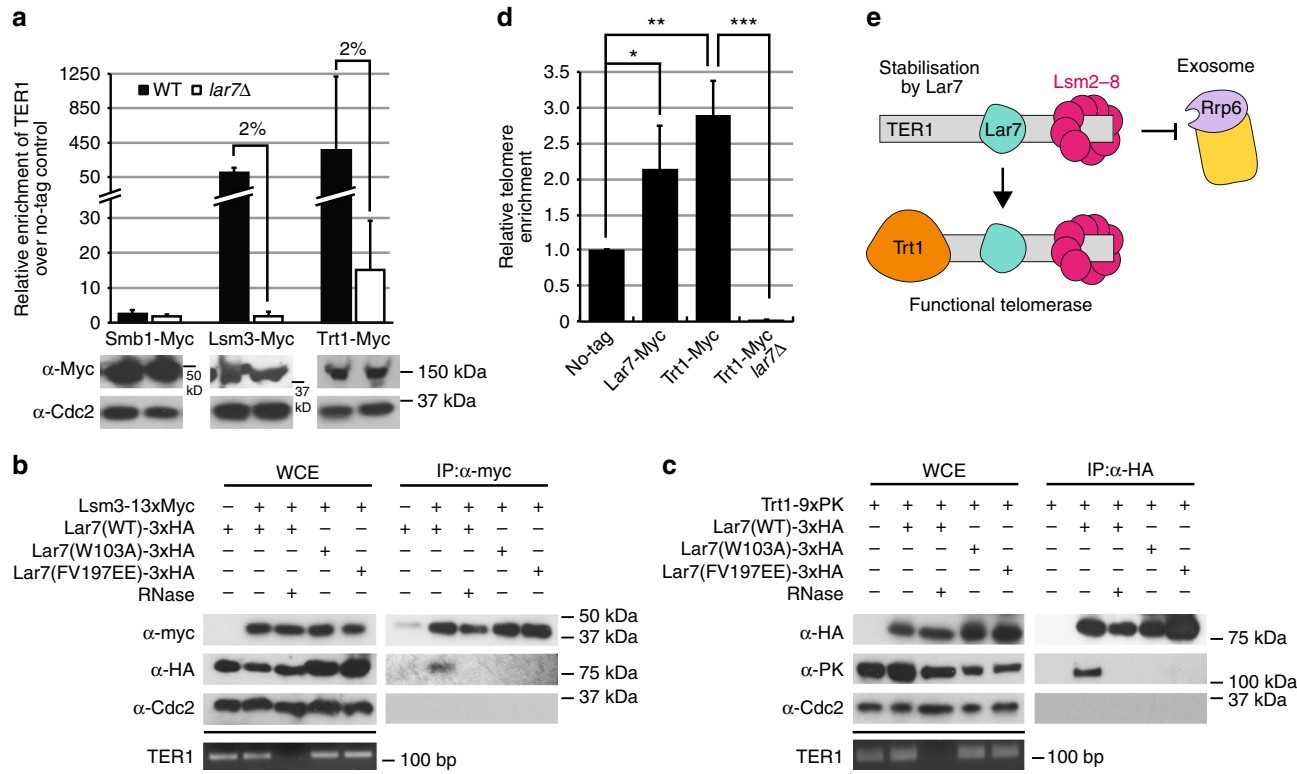

**Fig. 5** Lar7 is required for telomerase assembly and is an integral component of the telomerase complex. **a** Quantification using RT-qPCR of TER1 mRNA in lysates following immunoprecipitation using α-myc. Values were normalised first to actin and then to an untagged control. Data represented as a mean of two independent experiments. Error bars show standard deviation. Western blot demonstrates successful expression and immunoprecipiation. Cdc2 was used to control for sample input. **b** Whole-cell extracts (WCEs) were immunoprecipitated using α-myc in the presence or absence of RNase, as indicated. Resulting immunoprecipitates were hybridised with α-myc and α-HA. Cdc2 was used as a control for sample input. **c** Whole-cell extracts (WCE) were immunoprecipitated using α-HA. **d** Chromatin immunoprecipitation (ChIP) and qPCR to quantify telomeric DNA in lysates following immunoprecipitation with α-myc. Data is represented as a mean of three independent experiments. Error bars show standard deviation. Lar7-Myc and Trt1-Myc precipitated significantly more telomeric DNA vs. WT ($p = 0.0327$ and $p = 0.0024$, respectively). This was significantly abrogated in Trt1-Myc *lar7Δ* cells ($p \leq 0.0001$, unpaired $t$ test * at $p > 0.05$, ** at $p < 0.01$ and *** at $p < 0.001$). **e** Model indicating the role of Lar7 in stabilising the interaction between Lsm2–8 and TER1 and promoting functional telomerase assembly

functional telomerase. We surmise a model in which Lar7 functions in the TER1 biogenesis pathway, binding to stabilise the interaction with the Lsm2–8 complex and facilitating assembly with Trt1 (Fig. 5e).

We report two distinct functions of Lar7 in fission yeast: the suppression of Rrp6-mediated degradation of TER1 and the assembly of TER1 with LSm and Trt1 to form active telomerase. The Lar7 orthologue, *Tetrahymena* p65, is required for telomerase assembly as well as accumulation of telomerase RNA[37,42,43]. In addition, p65 is also required for expression of the catalytic subunit, which was not observed for fission yeast Lar7. In humans, the data set from a study using enhanced UV-crosslinking and immunoprecipitation (eCLIP)[44] suggested a weak or transient interaction may occur between hLARP7 and TERC. This may serve to explain the previously reported low telomerase activity and impaired telomere maintenance in LARP7 knockdown cells[30]. Thus, we believe that the LARP7-familiy proteins have a conserved function in telomerase biogenesis, in yeast, ciliates and potentially also in vertebrate species. As such, hLARP7 may also be a candidate disease gene in premature ageing syndromes such as dyskeratosis congenita.

Although the molecular mechanisms of the telomerase RNA maturation differ between species, two structural features of the RNA appear to be conserved: the template and pseudoknot (T-PK) domain and the stem-terminus element (STE, or the three-way junction domain)[45,46]. The telomerase catalytic subunit binds both the pseudoknot domain and the STE domain, which is required for its reverse-transcription activity. *Tetrahymena* p65 binds to the stem of the STE domain and promotes assembly of the catalytic subunit with both the T-PK and STE domains by facilitating a conformational change[37,43]. As Lar7 promotes TER1 interaction with Trt1 without directly binding it, we predict that Lar7 also facilitates a conformational change in TER1 that promotes telomerase assembly (Supplementary Fig. 6). The TER1 STE domain is essential for telomerase activity in *S. pombe*[46,47], and may therefore be the binding site for Lar7.

In mammals, TERC is protected from ribonuclease-dependent degradation via the binding of the H/ACA ribonucleoprotein complex (dyskerin, NOP10, NHP2 and GAR1). However, dyskerin mutants that prevent telomerase activity can be rescued by stabilisation of TERC[48], implying that, although protective, the human H/ACA complex is not directly involved in promoting the interaction of TERT and TERC. TCAB1 associates with the H/ACA complex and is required for trafficking of the active telomerase to the Cajal body and telomeres[13]. However, loss of TCAB1 does not impair telomerase activity or stability of TERC. Therefore, these factors are unlikely to be directly involved in telomerase assembly. Whether or not hLARP7 binds TERC and promotes a conformational change to facilitate assembly of TERT with the T-PK and STE domains remains elusive (Supplementary Fig. 6).

Collectively, we have shown that fission yeast Lar7 is a crucial factor in telomerase biogenesis and have identified a missing step

in the progression of telomerase RNA from precursor to stable mature RNP. Our findings demonstrate that, to a degree, assembly of the telomerase complex may be conserved between species and LARP7 proteins are a new factor universally involved in telomere maintenance. Considering the importance of Lar7 in *S. pombe*, further investigation is required to fully elucidate the telomeric role of hLARP7 in humans.

## Methods

**Media and growth condition**. Fission yeast strains used for this study are listed in Supplementary Table 1. Cells were cultured in standard rich medium (YES: FORMEDIUM™) at 32 °C. For TER1 overexpression assays, cells were cultured in synthetic minimal media (EMM: FORMEDIUM™).

**Strain construction**. The mutation and epitope fusion of the *lar7* gene were generated by replacement of the endogenous gene with the *lar7* replacing plasmids, pLar7c2-CheHA3. The *lar7* gene including 289 bases of the upstream region was cloned into the C-terminal tagging pNX3 vector (pNX3c2-mCherry-HA3[49]). The resulting plasmid was named pLar7c2-CheHA3. The cloned *lar7* gene from the plasmid was mutated using PCR-based methods and the Quickchange Lightning site-directed mutagenesis kit (Agilent Technologies). The endogenous *lar7* gene was replaced with the *kanMX6* cassette using PCR-based gene targeting method[49], to generate a strain carrying *lar7::kanMX6*. The *lar7* plasmid is digested with *Mfe*I and *Eco*RI to generate the *lar7* targeting fragment. A strain carrying the *lar7::kanMX6* allele was transformed with the digested pLar7c2 plasmid to replace the kanMX6 cassette with the *lar7-3xHA:natCX* fragment using previously described strategy[50]. A diploid strain carrying the *lar7::kanMX6* allele was transformed with the digested pLar7c2 plasmid. Whereas the cloned *lar7* promoter region is homologous to both allele of lar7, the terminator of the natCX cassette is only homologous to the deleted allele with that of the kanMX6 cassette. Therefore, the lar7-mCherry-3xHA:natCX fragment replaces the *lar7::kanMX6* allele[50]. Expression of wild-type and mutant Pof8 in newly generated strains was confirmed by western blot with anti-HA antibody (Supplementary Fig. 4). To express TER1 by the thiamine repressible nmt41 promoter, a pTeb1b-Pnmt41 plasmid was generated. The *ter1* gene including 299 bases of the upstream region and 429 bases of the downstream region were cloned into pNX3b vector[49]. The *nmt41* promoter was inserted at the transcription start site to generate pTer1b-Pnmt41. The endogenous *ter1* gene was replaced with the TKnatAX cassette using PCR-based gene targeting method[49]. The pTer1b-Pnmt41 was digested with *Cla*I and *Pme*I to generate the $P^{nmt41}$>*ter1* targeting fragment. A strain carrying the *ter1::TKnatAX* allele was transformed to replace the TKnatAX cassette with the $P^{nmt41}$>*ter1:hygMX6* fragment. Strains carrying Trt1-9xPK and 12xPK-Est1 and other deletion mutants are described before[20]. All other strains were generated by genetic crossing or PCR-based gene targeting.

**Uncropped images**. Uncropped versions of the western blot and DNA gel images presented in this manuscript can be found in Supplementary Figs. 7 and 8.

**Southern blot**. Southern blots to analyse telomere length were carried out on genomic DNA prepared from strains cultured >2 weeks after generation, unless indicated. Genomic DNA was quantified and normalised, and 40 µg digested with *Eco*RI. The digested DNA fragments were separated on 1% agarose gel and transferred by capillary action onto a nitrocellulose membrane. The membrane was probed overnight with a hot labelled 500 base telomere probe[20], before washing and exposure to a phosphor screen for 24 h.

**Protein extraction and immunoprecipitation (yeast)**. Logarithmically growing cells were collected and frozen at −80 °C. Pellets were resuspended in the same volume of HB buffer (50 mM HEPES/KOH at pH 7.5, 140 mM NaCl, 15 mM EGTA, 15 mM MgCl₂, 0.1% NP-40, 0.5 mM Na₃VO₄) containing protease inhibitors (1 mM dithiothreitol, 1 mM PMSF, 0.1% protein inhibitor cocktail set III (Sigma), 0.1 ng mL$^{-1}$ MG132 (Sigma), 10 U mL$^{-1}$ TURBO DNase (Ambion)). Cells were broken using a Fast Prep machine (Thermo), briefly sonicated using the Bioruptor and centrifuged to collect the chromatin containing cell extract. Monoclonal anti-Myc 9E11 (2276, Cell Signalling) and anti-HA antibodies (901514, Covance/BioLegend) were used for pull down and detection of Myc and HA tagged proteins, respectively. Antibodies were conjugated with mouse IgG-coated Dynabeads (Life Technologies) overnight. Whole-cell extracts (WCEs) were incubated with the pre-coated beads for 1 h at 4 °C, washed, and then resuspended in SDS loading buffer and boiled. Samples were subjected to western blotting with anti-HA (1:1000), anti-Myc (1:5000), or anti-PK (1:4000, anti-V5 SV5-Pk2, MCA2892, Covance/Bio-Rad) antibodies. Anti-Cdc2 (sc-53, Santa Cruz) was used as a control (1:5000).

**Reverse-transcription quantitative PCR**. RNA was collected either from logarithmically growing cells or from protein lysates following immunoprecipitation using Fungal RNA MiniPrep kit, Zymo Research. Genomic DNA was removed by

treatment with 10 U mL$^{-1}$ RNase-free TURBO DNase (Ambion). RNA of 1 µg was reverse-transcribed using 200U SuperScript IV reverse transcriptase (Thermo-Fisher Scientific) in a 20 µL reaction using 50 µM random hexamer oligos, according to the manufacturers guidelines. RT products were diluted to 200 µL with DEPC treated water and RNA was removed with RNase A. Samples were stored at −80 °C between use. RNA transcription was quantified using real-time PCR in a 20 µL reaction with 5 µL RT product, 2× SYBR Green PCR Master Mix (Roche) and 5 µM of each oligo, according to manufacturers guidelines. Reactions were run on a LightCycler 480 (Roche) under the following conditions: 95 °C for 10 min followed by 40 cycles of 10 s at 95 °C, 20 s at 60 °C and 10 s at 72 °C. Melting curve analysis was performed immediately after. Primer sets were TER1-preF1270 5′-AATTGCGTATTTAGTAAGAACGCG-3′ and TER1preR 5′-GATT-CATCACTTTCTCAAAATTTTGAAACCG-3′ for precursor TER1, TER1qPCRF 5′-CAGTGTACGTGAGTCTTCTGCCTT-3′ and TER1qPCRR 5′-CAAAA ATTCGTTGTGATCTGACAAGC-3′ for whole TER1 and Act1top 5′-GGATT-CCTACGTTGGTGATGA-3′ and Act1R277 5′-CGTTGTAGAAAGTGTG ATGCC-3′ for actin, snU1F5 5′-ACCTGGCATGAGTTTCTGC-3′ and snU1R105 5′-GACCTT-AGCCAGTCCACAGTTA-3′ for U1, and snU6F9 5′-GATCTTCG-GATCACTTTGGTC-3′ and snU6R151 5′-GGTTTTCTCTCAATGTCGCAG-3′ for U6. The level of TER1, U1 and U6 were normalised to actin mRNA, and TERC and 7SK were normalised to GAPDH mRNA using the ΔCT method. All data are expressed as ΔΔCT compared with control (wild type), unless otherwise stated. Average was calculated from biological replicas and unpaired *t* test analyses were performed to determine significance (* at $p < 0.05$, ** at $p < 0.01$ and *** at $p < 0.001$).

**Chromatin immunoprecipitation**. Cell fixation and immunoprecipitation were performed as previously described[51]. Cells were fixed in 1% formaldehyde solution (1% formaldehyde, 0.1 mM EDTA, 0.05 mM EGTA, 10 mM NaCl, 5 mM Tris-HCl pH8.0) for 15 minutes at 25 °C. Cells were resuspended in ice-cold ChIP-buffer1 (50 mM HEPES-KOH pH7.5, 140 mM NaCl, 1 mM EDTA, 1% Triton X-100, 0.1% Na deoxycholate). The Dynabeads conjugated anti-Myc 9E11 antibody was used for pull down of Myc tagged Trt1[51]. Precipitated DNA was amplified by qPCR and enrichment of telomeric DNA was calculated after normalising to the *act1* gene as an internal control. The DNA-binding efficiency is shown relative to the value obtained for Trt1-13xMyc or Lar7-13xMyc. Telomeric primers used for qPCR were 5′-CGGCTGACGGGTGGGGCCCAATA-3′ and 5′-GTGTGGAATTGA GTATGGTGAA-3′.

**Data availability**. The authors declare that (the/all other) data supporting the findings of this study are available within the paper (and its Supplementary Information files) or from the authors upon reasonable request.

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

## Acknowledgements

The fission yeast exo-ribonuclease strain collection and *rrp6Δ* strain were gifted from J. Bähler (University College London). The data extraction from the LARP7-eCLIP data set and analysis of the interaction of LARP7 with TERC were conducted by J. Ambrose, R. Jenner and J. Herrero (UCL Cancer Institute). This work is supported by Cancer Research UK (C36439/A12097) and the European Research Council (281722-HRMCB). Q.Y. is supported by the MSc Cancer at the UCL Cancer Institute. S.í.K. and A.A. are supported by the MSc Human Molecular Genetics program at the Imperial College London.

## Author contributions

K.T. and L.C.C. were responsible for study design, experimental work, data analysis, in silico analysis and manuscript preparation. K.T., L.C.C., S.í.K., Q.Y., V.M., C.P., A.A., H.A. S.R.P. and C.A.A. contributed to material construction, generation of preliminary results and reproducibility of data. L.C.C. performed RNA-IP, and L.C.C., T.L.W. and Q.Y. performed quantification of RNA stability. T.G. performed ChIP experiments. L.C.C. and C.P. analysed protein stability of telomeric proteins and telomerase components.

## Additional information

**Competing interests:** The authors declare no competing financial interests.

