## [Peer Review File · Nature Communications]

Reviewers' comments:

Reviewer #1 (Remarks to the Author):

The paper by Collopy et al describes the role of spPof8 and hLar7 in the telomerase assembly pathway. The paper is well written, interesting and an important contribution to the field. Briefly, Collopy et al. use sequence homology to identify spPof8 as an RNA binding protein containing a La-motive and an orthologue of p65. Based on this the authors rename spPof8 to spLar7. The authors then demonstrate that spLar7 is critical for telomere maintenance. spLar7 deletion and mutations lead to short telomeres, a defect in telomerase assembly and telomerase activity. In the last part of the paper the authors investigate the role of hLar7 showing that hLar7 leads to an assembly defect without effecting the levels of TR. While some minor points need clarification, the work is carefully and well done and I think of interest to the readers of Cell reports.

Comments:

- The telomere length differences in figure 2b are not very clear. It seems as if the tagged wild type Lar7 already has shorter telomeres. Can the authors provide a better gel or a quantification of the telomere length?
- Figure 2c does not seem to be normalized for the input of TER1 or assuming that TER1 levels are the same in all mutants. Although this is somewhat resolved later in figure 3 I found this very confusing at first.
- Figure 3: I am not sure why the author call the TER1 form that is detected with the common primer "mature". Should this be labeled "total" TER1? What primers are used in Figure 4a to quantify TER1? Is there a difference between mature and total? What do the results in Figure 3 and 4 look like if a primer is used that is specific to the mature form?
- Figure 5e: I find the label "to the telomere" confusing, as this somehow implies that there is a recruitment function somewhere. I think that the label "functional telomerase" suffices.
- Considering that the role of human Lar7 has been investigated previously, but with different outcomes I feel that the data shown in figure 6 are an add-on. I think understanding the role of hLar7 is very interesting and important to report, but the presented data are not sufficiently developed to correct the previous finding. The authors should consider developing this story more for a later publication.
- What antibody was used for the TERT blot in figure 6b. How was this validated?

Reviewer #2 (Remarks to the Author):

In the two dual-submission manuscripts by Paez-Moscoso et al and Collopy et al, the groups identify a new key member of the telomerase enzyme in fission yeast, termed respectively Pof8 and Lar7. Collopy et al first identify Pof8, a putative f-box protein, from a previous study (Liu et al) as a cause of a very short telomere phenotype. Because of domain similarity to the LARP7 protein family, Collopy et al rename the protein to Lar7. Paez_Moscoso et al first identify Pof8 while performing immunoprecipitation experiments from Lsm proteins (results in preparation for additional manuscript). They too note its similarities to LARP7 protein family. In fission yeast, the RNA component TER1 is generated initially by an incomplete spliceosomal cleavage reaction. It is first bound by Sm proteins and later by Lsm proteins and Trt1, the catalytic subunit.

To show that a protein is a core component of the telomerase enzyme, the authors should demonstrate that loss of the protein impairs telomere maintenance. They should also show that this protein can precipitate the other core telomerase components including the RNA component (TER1) and be able to precipitate a majority of telomerase activity. They should also investigate

the effect of loss of the protein on stability of the telomerase enzyme and its components and the effect on telomerase activity. Finally, they should look at which components of the telomerase complex this new protein is binding to.

Collopy: In this manuscript from Kazunori Tomita's lab, the authors find that pof8/lar7 is required for telomere maintenance, although telomeres can be maintained at a shorter length in the absence of pof8/lar7 by a telomerase dependent mechanism. They show that they can make targeted mutations in RNA-interacting domains of pof8/lar7 that impair telomere length maintenance and interaction of pof8/lar7 with TER1. They show that loss of pof8/lar7 impairs levels of mature ter1 but does not change precursor levels; this is dependent upon the RNA exosome. They show that lar7 interacts with lsm proteins in an RNA dependent manner; the same is true for Lar7 and Trt1. They study a relative of pof8/lar7 in human cells lines.

Minor Concerns

- In figure 1b, some lanes of the southern appear to have no telomeres. The authors interpret this as the telomeres being lost. It would be helpful to have some other DNA locus as a loading control; in the paper from Baumann's group they use the rad16 locus to prove there is DNA in the lane. This is also a problem in 4b.

- In figure 5b, the myc and HA labels in the WB from WCE are swapped.

- TERT antibodies do not work well for direct western blot due to low TERT levels in human cells.

I'm concerned that this is a background band. In figure 6b, it would be good to have lysate from a TERT- cell line, or lysate in which TERT has been knocked down, as a control for antibody specificity.

Major concerns

- There are no telomerase activity assays in this paper; it's unclear what effect pof8/lar7 has on telomerase activity, or how much telomerase activity is associated with pof8/lar7

- The data on human LARP7 are thin. They find that LARP7 is not required for TERC stability, but that it is required for TERC assembly with HA-TERT. However, they don't do a northern blot for TERC, so it's unclear whether levels of mature or immature TERC are changed. Additionally, their HA-TERT data has very large error bars, even though it's technically statistically significant. IP-PCR results to assess TERT-TERC association has been historically unreliable – instead, IP-northern has been robust. This is a sufficiently important claim that they should do the HA-TERT IP northern to confirm this finding.

Response to Reviewers' comments:

We thank both reviewers for their positive comments and constructive suggestions to improve our manuscript. We hope we have improved figures where required and better explained the controls and methods used to generate data, where unclear. A key change in the manuscript is the addition of Figure 6d-e, co-immunoprecipitation assay demonstrating a reduced interaction between dyskerin and TERT, but not with TERC, upon LARP7-knockdown. Our new data confirm that LARP7 is required for the assembly of TERT and TERC but not for TERC stability. We believe that this strengthens our data on the role of human LARP7 in telomere maintenance.

Please find our response (blue) to the Reviewer's comments (written in bold) below. We have indicated the page number (P) and line number (L) to reference where changes have been made within the main text. These have also been highlighted in the manuscript.

Reviewer #1 (Remarks to the Author):

The paper by Collopy et al describes the role of spPof8 and hLar7 in the telomerase assembly pathway. The paper is well written, interesting and an important contribution to the field. Briefly, Collopy et al. use sequence homology to identify spPof8 as an RNA binding protein containing a La-motive and an orthologue of p65. Based on this the authors rename spPof8 to spLar7. The authors then demonstrate that spLar7 is critical to for telomere maintenance. spLar7 deletion and mutations lead to short telomeres, a defect in telomerase assembly and telomerase activity. In the last part of the paper the authors investigate the role of hLar7 showing that hLar7 leads to an assembly defect without effecting the levels of TR. While some minor point need clarification, the work is carefully and well done and I think of interest to the readers of Cell reports.

We thank Reviewer #1 for these kind comments and appreciate their support of our manuscript.

Comments:

- **The telomere length differences in figure 2b are not very clear. It seems as if the tagged wild type Lar7 already has shorter telomeres. Can the authors provide a better gel or a quantification of the telomere length?**

We note that the 3xHA tag on Lar7 causes a slight decrease in wild-type telomere length. However, we have performed the Southern blot in Figure 2b again and now include a clearer image to emphasise the short telomere phenotype of *lar7-W103A*, *F197EE* and *I1341EE* mutants, which is comparable to that of the *lar7Δ* strain. We hope the shortening of telomeres in Lar7 mutant strains is now clearer.

- **Figure 2c does not seem to be normalized for the input of TER1 or assuming that TER1 levels are the same in all mutants. Although this is somewhat resolved later in figure 3 I found this very confusing at first.**

The previous Figure 2c demonstrated the level of TER1 that co-precipitated with a constant amount of Lar7-HA. However, we did not normalize this data to the level of input TERC. We agree that expression level of TER1 is lower in the *lar7* mutants, as presented in Figure 3b. We thank Reviewer #1 for highlighting this oversight and we now present this differently to avoid confusion.

In the text, we have removed the subheading “Lar7 associates with telomerase RNA”, under which Figure 2c was initially described. We now include the data of Figure 2c as the new Figure 3c, under the renamed subheading: “Lar7 associates with telomerase RNA and prevents the exosomal degradation of TER1” (P7 L25-26). This allowed us first to present Figure 3b, demonstrating low levels of TER1 in Lar7 mutants. Then we show in Figure 3c that Lar7 has an impaired interaction with TER1, even when normalised for TER1 input. The corresponding paragraph is moved to (P8 L10-20). We hope these changes address the concern of Reviewer #1 and better present our data to show an impaired interaction between Lar7 mutants and TER1.

- **Figure 3: I am not sure why the author call the TER1 form that is detected with the common primer “mature”. Should this be labeled “total” TER1? What primers are used in Figure 4a to quantify TER1? Is there a difference between mature and total? What do the result in Figure 3 and 4 look like if a primer is used that is specific to the mature form?**

We agree that qPCR for “mature” TER1 is indeed “total” TER1 RNA. As the entire sequence of mature TER1 features in the N-terminus of precursor TER1 we are unable to specifically amplify the mature product. The oligos used to amplify “total TER1” predominantly amplify the mature form, as this is more abundant. However, we apologise as this was not clearly described in Figure 3; ‘mature’ TER1 should indeed be ‘total TER1’ and we have re-labelled this in Figure 3. Accordingly, we have adjusted the main text to refer to ‘total’ rather than ‘mature’ TER1 where appropriate.

In Figure 4 we amplified total TER1, as data from Figure 3 showed that *lar7*Δ impacted only total TER1 levels, not precursor alone. We therefore inferred that the change in total RNA detected was a result of a reduction in the mature product: “This revealed a similar level of precursor RNA between *lar7*Δ cells and wild-type, but a significant reduction in the total product in the absence of Lar7 (Fig. 3a), suggesting low expression of the mature form.” (P8 L3-6).

- **Figure 5e: I find the label:” to the telomere” confusing, as this somehow implies that there is a recruitment function somewhere. I think that the label” functional telomerase suffices.**

We agree with Reviewer #1 that a recruitment function for Lar7 should not be implied. We previously stated 'to the telomere' to emphasize the result of our ChIP experiment (Figure 5d), which indicated Lar7 localizes to the telomere and, in *lar7Δ* cells, a significant reduction of telomerase at the telomere is seen. However, this is presumably caused by a defect in telomerase assembly and not by an impaired recruitment process. Hence, we have removed the appropriate part and simply stated 'functional telomerase' in Figure 5e, as advised.

- **Considering that the role of human Lar7 has been investigated previously, but with different outcomes I feel that the data shown in figure 6 are an add-on. I think understanding the role of hLar7 is very interesting and important to report, but the presented data are not sufficiently developed to correct the previous finding. The authors should consider developing this story more for a later publication.**

We thank Reviewer #1 for this advice and agree that the human LARP7 work, detailed in Figure 6, would benefit from further development to strengthen the statement made. We did not wish to exclude this data altogether as, like Reviewer #1, we believe it is an interesting and important finding that it highlights a function in LARP7 conserved to humans. We therefore provide additional data, shown in Figure 6d-f. We performed a co-immunoprecipitation experiment that detected a reduced interaction between TERT and dyskerin upon LARP7-knockdown (Figure 6d). We show in Figure 6e that, in LARP7-knockdown cells, dyskerin still interacts well with TERC. From this we therefore confirmed that telomerase assembly is impaired in the absence of LARP7; although TERC is stabilized by dyskerin, LARP7 is required for the assembly of the TERC-TERT ribonucleoprotein complex. A schematic summarizing this data is now shown in Figure 6f. We therefore hope this strengthens our initial finding that telomerase assembly is impaired by the knockdown of LARP7.

- **What antibody was used for the TERT blot in figure 6b. How was this validated?**

We report in our methods section that the antibody used was anti-telomerase reverse transcriptase [Y182] from Abcam (ab32020). We appreciate that endogenous telomerase is difficult to detect due to low expression, however with a longer exposure we were able to get a clear signal. The antibody used has been previously validated (Xi *et al.* 2014), however we have now included our own analysis as a new Figure S5. We hope this reinforces the result shown in Figure 6b.

Reviewer #2 (Remarks to the Author):

In the two dual-submission manuscripts by Paez-Moscoso et al and Collopy et al, the groups identify a new key member of the telomerase enzyme in fission yeast, termed respectively Pof8 and Lar7. Collopy et al first identify Pof8, a putative f-box protein, from a previous study (Liu et al) as a cause of a very short telomere phenotype. Because of domain similarity to the LARP7

protein family, Collopy et al rename the protein to Lar7. Paez_Moscoso et al first identify Pof8 while performing immunoprecipitation experiments from Lsm proteins (results in preparation for additional manuscript). They too note its similarities to LARP7 protein family.

In fission yeast, the RNA component TER1 is generated initially by an incomplete spliceosomal cleavage reaction. It is first bound by Sm proteins and later by Lsm proteins and Trt1, the catalytic subunit.

To show that a protein is a core component of the telomerase enzyme, the authors should demonstrate that loss of the protein impairs telomere maintenance. They should also show that this protein can precipitate the other core telomerase components including the RNA component (TER1) and be able to precipitate a majority of telomerase activity. They should also investigate the effect of loss of the protein on stability of the telomerase enzyme and its components and the effect on telomerase activity. Finally, they should look at which components of the telomerase complex this new protein is binding to.

Collopy: In this manuscript from Kazunori Tomita's lab, the authors find that pof8/lar7 is required for telomere maintenance, although telomeres can be maintained at a shorter length in the absence of pof8/lar7 by a telomerase dependent mechanism. They show that they can make targeted mutations in RNA-interacting domains of pof8/lar7 that impair telomere length maintenance and interaction of pof8/lar7 with TER1. They show that loss of pof8/lar7 impairs levels of mature ter1 but does not change precursor levels; this is dependent upon the RNA exosome. They show that lar7 interacts with lsm proteins in an RNA dependent manner; the same is true for Lar7 and Trt1. They study a relative of pof8/lar7 in human cells lines.

We thank Reviewer #2 for their comments and hope our manuscript reports the key findings suggested.

Minor Concerns

- In figure 1b, some lanes of the southern appear to have no telomeres. The authors interpret this as the telomeres being lost. It would be helpful to have some other DNA locus as a loading control; in the paper from Baumann's group they use the rad16 locus to prove there is DNA in the lane. This is also a problem in 4b.

We apologise for this oversight and have now included images of a different DNA locus in Figures 1b and 4b to demonstrate an even loading of DNA in each lane.

- **In figure 5b, the myc and HA labels in the WB from WCE are swapped.**

We thank Reviewer #2 for spotting this error and have rectified the labeling of Figure 5b.

- **TERT antibodies do not work well for direct western blot due to low TERT levels in human cells. I'm concerned that this is a background band. In figure 6b, it would be good to have lysate from a TERT- cell line, or lysate in which TERT has been knocked down, as a control for antibody specificity.**

We appreciate the concern of Reviewer #1 and #2 on this matter and we are aware of the difficulties detecting endogenous TERT in human cells. As described for Reviewer #1 above, we have now added included Figure S5, which demonstrates the success of the TERT antibody used. We show a band appearing for a control cell extract of approximately 120 kDa, a similar size to 3xHA-TERT over-expressed using a pCDNA-3xHA-TERT construct. This band is diminished up on transfection with siRNA against hTERT RNA.

Major concerns

- **There are no telomerase activity assays in this paper; it's unclear what effect *pof8/lar7* has on telomerase activity, or how much telomerase activity is associated with *pof8/lar7***

In this manuscript, we demonstrated that telomerase activity is low *in vivo* in *lar7Δ* cells. This was caused by impaired association of Trt1 with TER1. Localisation of Lar7 at telomere suggests that Lar7 associates with mature telomerase. Since telomerase activity essentially requires complex formation of Trt1 and TER1, we felt that this was a sufficient demonstration of the requirement of Lar7 for normal telomerase performance. Hence, we feel that a telomerase activity assay would not provide any further insight into this. We hope that our data, submitted in conjunction with Peter Baumann's lab, provides satisfactory evidence to support the requirement of Lar7 for telomerase biogenesis and therefore normal telomerase activity.

- **The data on human LARP7 are thin. They find that LARP7 is not required for TERC stability, but that it is required for TERC assembly with HA-TERT. However, they don't do a northern blot for TERC, so it's unclear whether levels of mature or immature TERC are changed. Additionally, their HA-TERT data has very large error bars, even though its technically statistically significant. IP-PCR results to assess TERT-TERC association has been historically unreliable – instead, IP-northern has been robust. This is a sufficiently important claim that they should do the HA-TERT IP northern to confirm this finding.**

We thank Reviewer #2 for their advice on this matter. We agree that our IP-PCR data demonstrates a statistically significant reduction in the interaction between TERT and TERC

in LARP7-depleted cells (Figure 6c, $p \leq 0.01$). We had attempted to perform an IP-Northern instead, however the quantity of RNA acquired from the IP is not sufficient for detection using Northern blotting. An experiment with pcDNA+TERC plasmid suggested that at least 1 ng of TERC is required for a detectable signal using the 'dot blot' method, and less than this level is recovered following TERT immunoprecipitation and RNA extraction. For this reason we performed qPCR. The RNA-IP-qPCR represents an average. Every experiment resulted in impaired TERC pull down after LARP7-knock down. The variability of the qPCR result reflects the variability of RNA-IP efficiency and the inefficiency of siRNA knockdown of LARP7 (Figure 6b).

In order to strengthen and complement our finding in Figure 6c, we now include additional data. We have demonstrated *via* co-IP that LARP7-knockdown impairs the interaction between TERT and dyskerin (Figure 6d). However, we show that dyskerin is still able to complex with TERC in the absence of LARP7 (Figure 6e). Dyskerin binds to only mature TERC (polyA tail trimmed). Hence, the majority of TERC is mature and stable in the cell as it is bound to dyskerin. Importantly, LARP7-knockdown results in impaired assembly between TERT and the TERC-dyskerin complex, confirming the result in Figure 6c. We summarise this with a schematic (Figure 6f). This new data is described in the manuscript as follows:

“To validate this result further, we examined the impact of LARP7 loss on the interaction between TERT and dyskerin. In wild-type cells, TERT co-purified with dyskerin, as both are contained in the mature telomerase complex. However, this interaction was lost upon LARP7-knockdown (Fig. 6d). Levels of TERC RNA extracted following immunoprecipitation were measured by qPCR, and indicate that LARP7-loss does not impair the interaction between dyskerin and TERC (Fig. 6e). This serves to explain the unaltered stability of TERC detected in the absence of LARP7 (Fig. 6b). We therefore conclude that, in the absence of LARP7, TERC is stabilised by dyskerin binding but unable to complex with TERT (Fig 6f).” (P11, L10-18).

The methods section has been updated to detail the protocol used (P17, L20-27) and the descriptions of the experiments performed explained in the figure legend (Figure 6d-f). We believe that the claims we have made are important and now hope that this new data reports this in a more robust fashion.

REVIEWERS' COMMENTS:

Reviewer #1 (Remarks to the Author):

The authors have addressed my concerns. I think this paper is ready for publication in Nature communications.

Reviewer #2 (Remarks to the Author):

The data on Pombe telomerase are convincing and should be published. The data in Figure 6 on human telomerase remain unconvincing. There doesn't seem to be any rationale for human LARP7 being involved in human telomerase, such as binding with a telomerase component. It has been very challenging to establish proteins as being required for telomerase function - such a statement would require much more extensive studies. The Tert westerns remain unconvincing - the IP dyskerin blots for Tert show bands that have different migration patterns. The IP 3xHA-TERT coupled with qRT PCR assay is insufficient to support the idea that there is less TERC loaded with overexpressed TERT. These data would subtract from this otherwise excellent study. The authors should remove Fig. 6, edit the paper to focus on the Pombe data and invest additional effort to understand if hLARP7 has any role in human telomerase.

Response to Reviewers' comments:

We thank the reviewers for their comments on our manuscript. As Reviewer #1 was satisfied with the changes made during the first revision, we have focussed on the suggestion of Reviewer #2 below. We now hope our manuscript is ready for publication.

Reviewer #2

The data on Pombe telomerase are convincing and should be published. The data in Figure 6 on human telomerase remain unconvincing. There doesn't seem to be any rationale for human LARP7 being involved in human telomerase, such as binding with a telomerase component. It has been very challenging to establish proteins as being required for telomerase function - such a statement would require much more extensive studies. The Tert westerns remain unconvincing - the IP dyskerin blots for Tert show bands that have different migration patterns. The IP 3xHA-TERT coupled with qRT PCR assay is insufficient to support the idea that there is less TERC loaded with overexpressed TERT. These data would subtract from this otherwise excellent study. The authors should remove Fig. 6, edit the paper to focus on the Pombe data and invest additional effort to understand if hLARP7 has any role in human telomerase.

We thank Reviewer #2 for their comments on our study and are sorry that they remain unconvinced by the human data. To address this, we have followed their advice and removed Figure 6 and edited the paper to remove all human findings, focussing on the *S. pombe* data. The changes made are as follows:

- We have deleted the following statement from the abstract:

‘In humans, siRNA-mediated depletion of hLARP7 did not impair the stability of telomerase components. However, we observed that hLARP7 promotes the interaction between telomerase RNA and reverse transcriptase, similarly to Lar7 in fission yeast.’
- We have also added the following statement to the abstract to emphasise the findings in fission yeast:

‘Lar7 remains a component of the mature telomerase complex and is required for telomerase localisation to the telomere.’ (P2L10-12)
- From the introduction we have removed the summary finding:
‘We also show that hLARP7 knockdown results in an impaired interaction between telomerase RNA and reverse transcriptase, similarly to loss of Lar7 in fission yeast.’
- We have deleted the results section, describing the data in Figure 6, entitled ‘Human LARP7 functions in telomerase complex assembly’. We have also removed the legend to Figure 6 and relevant methods and present the revised manuscript without Figure 6. We have also removed Figure S5, as this validated the TERT antibody used and is no longer required. Accordingly, Figure S6 has been relabelled S5.
- We have removed the following statements from the discussion section:

‘In humans, we demonstrate that hLARP7 facilitates the association of TERC with TERT (but not dyskerin), highlighting a partially conserved function of the LARP7 family in telomerase assembly.’

‘...although a stable interaction of hLARP7 with TERC was not detected in our assay’

‘This may well be the case, as hLARP7 is not required for TERC stability or dyskerin interaction, however it is required for a stable TERC-TERT association, akin to Lar7 and p65.’

‘Indeed, human LARP7 is also involved in telomerase assembly and telomere length homeostasis.’

- The edited section of the discussion now reads:

‘In humans, the dataset from a study using enhanced UV-crosslinking and immunoprecipitation (eCLIP)⁴⁴ suggested a weak or transient interaction may occur between hLARP7 and TERC. This may serve to explain the previously reported low telomerase activity and impaired telomere maintenance in LARP7 knockdown cells³⁰. Thus, we believe that the LARP7-family proteins have a conserved function in telomerase biogenesis, in yeast, and potentially also in vertebrate species.’ (P10L27-P11L4)